# Antifungal Activity and Type of Interaction of *Melissa officinalis* Essential Oil with Antimycotics against Biofilms of Multidrug-Resistant *Candida* Isolates from Vulvovaginal Mucosa

**DOI:** 10.3390/jof9111080

**Published:** 2023-11-04

**Authors:** Marina Ranđelović, Marina Dimitrijević, Suzana Otašević, Ljiljana Stanojević, Milica Išljamović, Aleksandra Ignjatović, Valentina Arsić-Arsenijević, Zorica Stojanović-Radić

**Affiliations:** 1Department of Microbiology and Immunology, Medical Faculty, University of Nis, 18000 Nis, Serbia; otasevicsuzana@gmail.com; 2Centre of Microbiology, Public Health Institute Nis, 18000 Nis, Serbia; 3Department of Biology, Faculty of Science and Mathematics, University of Nis, 18000 Nis, Serbia; marina.dimitrijevic@pmf.edu.rs (M.D.); zorica.stojanovic-radic@pmf.edu.rs (Z.S.-R.); 4Department of Chemistry and Chemical Technology, Faculty of Technology, University of Nis, 18000 Nis, Serbia; jiljas76@yahoo.com; 5Department of Dental Health Care, Health Center Niš, 18000 Nis, Serbia; petrovicmilica21@gmail.com; 6Department of Medical Statistics and Informatics, Medical Faculty, University of Nis, 18000 Nis, Serbia; drsalea@yahoo.com; 7Department of Microbiology and Immunology, Medical Faculty, University of Belgrade, 11000 Belgrade, Serbia; mikomedlab@yahoo.com

**Keywords:** vulvovaginal candidosis, biofilm, *Melissa officinalis*, fluconazole, nystatin

## Abstract

(1) Background: Vulvovaginal candidosis (VVC) is a major therapy issue due to its high resistance rate and virulence factors such as the ability to form biofilms. The possibility of combining commonly used antifungals with natural products might greatly improve therapeutic success. (2) Methods: A total of 49 vulvovaginal isolates, causative agents of recurrent VVC, were tested for their susceptibility to fluconazole, nystatin, and *Melissa officinalis* essential oil (MOEO). This examination included testing the antibiofilm potential of antifungals and MOEO and the determination of their types of interaction with mature biofilms. (3) Results: Antimicrobial testing showed that 94.4% of the *Candida albicans* isolates and all the *Candida krusei* isolates were resistant to fluconazole, while all strains showed resistance to nystatin. The same strains were susceptible to MOEO in 0.156–2.5 mg/mL concentrations. Additionally, the results revealed very limited action of fluconazole, while nystatin and MOEO reduced the amount of biofilm formed by as much as 17.7% and 4.6%, respectively. Testing of the combined effect showed strain-specific synergistic action. Furthermore, the lower concentrations exhibited antagonistic effects even in cases where synergism was detected. (4) Conclusions: This study showed that MOEO had a very good antibiofilm effect. However, combining MOEO with antimycotics demonstrated that the type of action depended on the choice of antifungal drugs as well as the applied concentration.

## 1. Introduction

Vulvovaginal candidiasis (VVC) is a common condition in women, with prevalence rates ranging from 15% to nearly 40% [1]. Numerous studies have shown that 75% of women will undergo at least one episode of vulvovaginal candidiasis in their life, and 5–10% of all women will develop recurrent vulvovaginal candidiasis (RVVC), which is defined as at least four episodes of vulvovaginal candidiasis per year [2,3].

According to experimental findings, approximately 20% of asymptomatic women can be colonized by *Candida* (*C.*) species, predominantly *C. albicans* [4]. Despite this fact, even a modest alteration in host defense and the composition of the resident bacterial community can promote opportunistic infections with this yeast [5]. The host factors that can lead to VVC development are pregnancy, poorly controlled diabetes, hormonal disbalance immunosuppression, the use of broad-spectrum antibiotics and glucocorticoids, stress, allergies, and genetic predisposition [6,7,8]. Other reported factors include the use of intrauterine devices or oral contraceptives, estrogen therapy, and spermicides, as well as individual hygiene, clothing, and sexual behavior [9].

It is widely known that *C. albicans* is the major cause of RVVC [10]. In recent years, however, non-*albicans Candida* (NAC) species, such as *C. glabrata* or *C. krusei*, have been established as the cause of both sporadic and RVVC [11,12]. In addition, the published studies suggest that the yeast *Saccharomyces cerevisiae* (*S. cerevisiae*) may also cause genital fungal infections [13,14]. The problem of recurrence could be due to the pathogenic potential of the causative species, and many studies consider virulence factors to be associated with the ability to produce biofilms. It has been demonstrated that *Candida* spp. are capable of forming biofilms, which are defined as a formation consisting of a community of the cells of microorganisms that are protected from the immune system, show lower growth rates, exhibit decreased susceptibility to antimicrobials, and present a constant source of infection [15]. Various investigations have been conducted to characterize biofilm formation, pathogenicity, and antifungal susceptibility profiles in clinical isolates from VVC and RVVC patients [16,17,18]. It has been proven that the production of *Candida* biofilms influences the virulence and type of resistance to traditional treatments of VVC [19].

It has already been noted that NAC species are more resistant to antifungal medications, especially azoles, the first-line treatment for these infections, resulting in treatment failure [20]. Since *C. krusei* is resistant to fluconazole (FLU) and as the use of azoles for *C. glabrata* infections is frequently unsuccessful, topical nystatin (NY), amphotericin B, flucytosine ciclopiroxolamine, or boric acid, as well as combinations of these agents, are recommended for treating VVC caused by NAC species. However, few studies have examined the effectiveness and consequences of the long-term use of these drugs [21]. Additionally, increasing resistance of *C. albicans* to azoles has also been reported [22]. In such cases, using other azoles, such as oral itraconazole or ketoconazole and topical clotrimazole, is recommended if cross-resistance has not been established, although it has not been proven that their long-term use is safe [23,24].

Current research on treating VVC aims to establish new, effective antifungals for local or systemic application [18,25,26] but also to examine the possible synergistic effect of antifungals and natural substances on the biofilms formed by *Candida* strains [27,28,29,30].

Previous research on the antibiofilm activity of plants’ secondary metabolites in essential oils showed their significant impact in preventing the formation and inhibiting the growth of already-formed biofilms of bacteria and fungi [31,32]. The essential oil (EO) of the plant *Melissa officinalis*, a member of the *Lamiaceae* family, is widely used in traditional medicine due to its many benefits [33]. The antimicrobial activity of this EO has already been proven in numerous in vitro studies against a wide range of Gram-positive and Gram-negative bacteria as well as fungi [34,35,36].

However, only a few studies have examined the effect of *Melissa officinalis* EO (MOEO) on biofilm formation [37,38], and a study conducted by Raut et al. has shown that some components of this EO can affect mature *C. albicans* biofilms [39]. Nevertheless, the synergistic effect of MOEO with antifungals on the biofilm formed by *Candida* species has not been investigated so far.

Since EOs have already been used for the treatment of VVC [40,41], the aim of this study was to determine the antifungal activity of MOEO, FLU, and NY as well as examine the combined effect of MOEO with antimycotics against biofilms of *Candida* isolates, the causative agents of VVC.

## 2. Materials and Methods

### 2.1. Essential Oil

Commercially available *Melissa officinalis* essential oil produced by Promontis (Vilandrica, municipality Gadžin Han, Nišava District, Serbia) (lot: 8606110004031) was provided and stored at 4 °C until usage.

### 2.2. Gas Chromatography/Mass Spectrometry (GC/MS) and Gas Chromatography/Flame Ionization Detection (GC/FID) Analysis of Melissa officinalis Essential Oil

GC/MS analysis was performed using an Agilent Technologies 7890B gas chromatograph, equipped with a nonpolar, silica capillary column HP-5MS (5% diphenyl- and 95% dimethyl-polysiloxane, 30 m × 0.25 mm, 0.25 μm film thickness; Agilent Technologies, Santa Clara, CA, USA), and coupled with inert, selective 5977A mass detector produced by the same company. The EO obtained was dissolved in diethyl ether. One μL of the solution prepared was injected into the GC column through a split/splitless inlet set at 220 °C in 40:1 split mode. Helium was used as the carrier gas at a constant flow rate of 1 cm^3^/min. The oven temperature was increased from 60 °C to 246 °C at a rate of 3 °C/min. Temperatures of the MSD transfer line, ion source, and quadrupole mass analyzer were set at 300 °C, 230 °C, and 150 °C, respectively. The ionization voltage was 70 eV, and the mass range was *m*/*z* 41–415. GC/FID analysis was carried out under identical experimental conditions as those applied for GC/MS. The flows of the carrier gas (He), make up gas (N_2_), fuel gas (H_2_), and oxidizing gas (Air) were 1, 25, 30, and 400 cm^3^/min, respectively. The temperature of the flame-ionization detector (FID) was set at 300 °C. Data processing was performed using MSD ChemStation, MassHunter Qualitative Analysis and AMDIS_32 software (2.70) (Agilent Technologies, Santa Clara, CA, USA). Retention indices of the components from the analyzed samples were experimentally determined using a homologous series of n-alkanes from C8–C20 as standards. Essential oil constituents were identified based on a comparison of their retention indices (RI exp) with those available in the literature [42]—RI lit; their mass spectra were compared with those from the Willey 6, NIST2011, and RTLPEST3 libraries, and, whenever possible, an authentic standard was co-injected (Co-I). Content (%) of the compounds present in essential oil was determined using the area normalization method, without any corrections.

### 2.3. Isolation and Identification of Yeast Isolates

Forty-nine strains were obtained from the mycological laboratory of the Institute for Public Health Nis. They were identified as *C. albicans* and NAC species in a Candida-Chromogenic medium (Liofilchem/Bacteriology products, Roseto degli Abruzzi, Italy) and stored at −80 °C. Preliminary antimycotic susceptibility analysis showed that they belonged to the group of highly resistant strains; this finding was obtained using the Integral system yeast plus (ISYP) test (Liofilchem, Italy). Identification of the strains was performed using matrix-assisted laser desorption/ionization time-of-flight mass spectrometry (MALDI-TOF MS). Briefly, after 24 h incubation on SDA at 37 °C, a thin colony layer was created using a toothpick on a target plate with 96 spots (Bruker Daltonics, Bremen, Germany) so that one colony could be placed in one spot of the plate. As a positive control, a bacterial test standard (Bruker Daltonics, Bremen, Germany) was used. Samples were dried at room temperature and treated with 1 µL of 70% formic acid, and, a few minutes later, alpha-cyano-4-hydroxycinnamic acid (HCCA) matrix was added. After final drying, the plate was inserted into the MALDI-TOF MS Biotyper Sirius one IVD System (Bruker Daltonics, Bremen, Germany), and identification of isolates was performed using MBT Compass software, version 4.1.100, in automatic runs provided by flexControl, version 3.4.207.20 (Bruker Daltonics, Bremen, Germany). The MBT Compass Library, Revision H, 3893 species/entries database was used in this research to compare generated mass spectra. If the log score values were larger than 1.7, identification was considered valid. *C. albicans* ATCC 10259 was used as a control strain.

### 2.4. Antimicrobial Testing

#### 2.4.1. Susceptibility of Isolates to Antimycotics

Besides preliminary testing conducted via ISYP test, the susceptibility of the 49 strains to two widely used antimycotic agents, FLU and NY (Sigma-Aldrich Company, Steinheim, Germany), was tested using broth microdilution method with slight modifications [43]. Stock solutions for all antimycotics were prepared in dimethyl sulphoxide (DMSO) at concentrations of 2.560 µg/mL for FLU and 160 µg/mL for NY.

Double-diluted antimycotics were prepared in 96-well plastic microtiter plates before being transferred to new plates that had already been pre-filled with RPMI-1640 Medium (Sigma-Aldrich, Darmstadt, Germany) with L-glutamine and without bicarbonate. In the wells containing the antimycotics and RPMI, as well as in those designed to be the growth control wells (RPMI without antimycotics), the prepared suspensions of yeast strains adjusted to a density of 0.5 McFarland standard (obtained from colonies cultured on SDA for 24 h at 35 °C) were added. The final concentrations of the fungal suspensions were between 0.5 and 2.5 × 10^5^ CFU/mL. FLU concentrations ranged from 0.062 to 128 µg/mL, while NY values ranged from 0.008 to 16 µg/mL. For each strain, a microdilution test was performed in triplicate. In this assay, a vehicle control with only the solvent (DMSO) was established to exclude the possible effect of solvent activity. Incubation of the inoculated microplates was carried out at 37 °C for 24 h. Here, 100% growth inhibition for NY and 50% growth inhibition for FLU were used to define the minimum inhibitory concentration (MIC) endpoints [43]. The strains used for quality control were *C. parapsilosis* ATCC 22019 and *C. krusei* ATCC 6258.

#### 2.4.2. Susceptibility of Isolates to Essential Oil

The stock solution of MOEO was obtained by diluting the 10 mg of EO in 1 mL of dimethyl sulfoxide (DMSO). Afterward, this solution was serially diluted in RPMI 1640 medium (Sigma-Aldrich, Darmstadt, Germany) with L-glutamine and without bicarbonate so that the final concentrations of EO ranged from 0.078 mg/mL to 10 mg/mL. The rest of the procedure was the same as that applied when testing the sensitivity of the examined strains to antimycotics. Furthermore, 100% growth inhibition was defined as the MIC endpoint.

### 2.5. Biofilm Assays

#### 2.5.1. Quantification of Biofilm Biomass via CV Staining

Isolated and quality control strains were subcultured on SDA for 24 h, and from the obtained cultures, suspensions corresponding to 0.5 McFarland turbidity were created in sterile phosphate-buffered saline (PBS). Crystal violet (CV) assay was used to quantify produced biofilms in 96-well microtiter plates under static conditions [44,45]. Firstly, RPMI-1640 medium supplemented with 0.8% glucose was added to each well, and after adding the suspensions, the final cell concentrations were ~5 × 10^5^. Following 48 h of incubation at 35 °C, the contents of the well were carefully aspirated, washed three times with sterile phosphate-buffered saline (PBS, pH = 7.4), dried, and stained using 0.5% CV for 20 min. Testing was performed in triplicate. After removing excess dye, the remaining CV in biofilms was extracted by adding 250 µL of 96% (*v*/*v*) ethanol to each well for 45 min. After this step, 150 µL of the prepared solutions was transferred into a sterile microtiter plate. Monitoring of biofilm growth was conducted based on absorbance measurements at 595 nm using an ELISA reader (Multiskan™ FC Microplate Photometer, Thermo Scientific™, Waltham, MA, USA). The strains were classified into the following four categories based on their production ability, namely, non-producers and weak, moderate, and strong biofilm producers, according to Stepanović et al. [46].

#### 2.5.2. Evaluation of Biofilm Metabolic Activity via MTT Assay

The activities of each agent on the formed biofilm were tested using MTT (3-(4,5-dimethyl-2-thiazolyl)-2,5-diphenyl-2H tetrazolium bromide) assay. Eight strains, four *C. albicans* and four *C. glabrata* isolates selected according to their highest resistance rate to FLU, NY, and MOEO and high-biofilm-producing abilities, were examined. Also, the combined effects of FLU and NY with MOEO on biofilms’ growth and metabolic activity were examined. For all experiments, the biofilms had been previously formed in 96-well, flat-bottomed plates. In each well, 200 µL of the yeast cell suspension was added, which resulted in a final cell concentration of 1–5 × 106 cells/mL in RPMI-1640. After incubation at 35 °C for 48 h, biofilms were formed, and non-adherent cells were removed, resulting in pre-formed biofilms for further testing.

For the determination of the antibiofilm activity of MOEO, FLU, and NY, 200 µL of the drug suspension (EO or antimycotic) in RPMI was added at concentrations of 2 × MIC, 1 × MIC, and 0.5 × MIC and incubated at 35 °C for 24 h. The contents were removed, and PBS (pH 7.4) was used to wash the wells three times. Each well was then filled with 100 µL of MTT solution (5 mg/mL; Sigma-Aldrich Company, Steinheim, Germany). The plate was incubated for 4 h at 35 °C in the dark. After removing the supernatant, 100 µL of DMSO was added to each well, and the plate was incubated at 35 °C for 10 min and protected from the light. Subsequently, 80 µL of solvent was transferred from each well to another plate, and reading was performed at 570 nm. For every plate, growth and sterility controls were developed. Triplicates of each test were performed. Results are presented as the relationships of the absorbance readings between the treated wells and the controls (free of EO and antimycotic).

### 2.6. Checkerboard Assay

The checkerboard technique was used to evaluate the combined effect of EO and antimycotics on already-formed biofilms [47]. Based on the results obtained for the MICs of the planktonic cells for each strain, the synergistic potential of the combination of MOEO with FLU and NY on the *Candida* biofilm was evaluated.

The mature biofilm was prepared in the same way as described above. After removing the non-adherent cells, 150 µL of RPMI, 25 µL of serially diluted MOEO (horizontal), and 25 µL of serially diluted antimycotic (FLU in one plate and NY in another one) (vertical) were added to wells of a 96-well flat-bottomed plate so that final concentrations of both drugs tested were 0.0625 × MIC, 0.125 × MIC, 0.25 × MIC, 0,5 × MIC, 1 × MIC, and 2 × MIC (for each isolate). Growth (containing drug-free biofilm), sterility (RPMI-1640), and drug controls were included in each plate. The plates were incubated at 35 °C for 24 h, and after removing the content and washing the plates three times with PBS, 100 µL of MTT solution was added, and biofilms were further quantified as described above. Reduction in the amount of biofilm that was higher than 90% of the control growth was used as an endpoint for MIC determination.

### 2.7. Statistical Analysis

Antibiofilm efficacy experimental results are expressed as the mean ± SD, and statistically significant differences were determined using two-way analysis of variance (ANOVA) followed by Tukey’s post-hoc test for multiple comparisons (Graphpad Prism version 6). Probability values (*p*) less than 0.05 were considered statistically significant.

To model the interaction between MOEO and antifungal agents (FLU and NY), the response-surface analysis (RSA) method based on the Bliss independence model was applied [48]. The dose response surface in RSA was modeled using the dose–response curves of each antifungal agent (single-agent dose–response curves), which reflected the growth rates achieved in each well exposed to a single agent [47]. The response-surface analysis can also be implemented independent of an inhibition endpoint, the latter of which we decided should be 90% for our data set. The output of the response surface analysis of the Bliss-independence-based drug interaction model is presented through a three-dimensional shaded interaction surface and a synergy distribution matrix, where Bliss synergy scores (SBliss) for each concentration pair of antifungal agents are displayed. The RSA was conducted using Combenefit software (version 2.021) [48].

## 3. Results

### 3.1. Chemical Composition of Essential Oil

The chemical composition analysis allowed for the identification of 39 components, which comprised 99.8% of the total peak area presented in Figure 1. The most abundant group of compounds belonged to group of oxygen-containing monoterpenes (61.4%), followed by sesquiterpene hydrocarbons (29.9%), oxygen-containing sesquiterpenes (4.0%), monoterpene hydrocarbons (2.3%), and others (2.2%). The main constituents of the MOEO were geranial (31.0%), neral (19.7%), (E)-caryophyllene (19.4%), which structures are given in Figure 2, as well as germacrene D (5.3%), and citronellal (4.6%). The less common components included caryophyllene oxide (3.5%), 6-Methyl-5-hepten-2-one (2.2%), geranyl acetate (2.0%), α-Humulene (1.4%), linalool and δ-Cadinene (1.3% each), α-copaene, and €-β-Ocimene (1.2% each), while the other components were represented with an amount of less than 1% (Table 1).

### 3.2. Broth Microdilution Assay

Susceptibility to FLU, NY, and MOEO, as well as biofilm production ability, was tested for 49 strains (18 *C. albicans*, 17 *C. glabrata*, 11 *C. krusei*, 3 *S. cerevisiae*) that showed the highest resistance rates after being tested using the ISYP test (Table 2). The susceptibility breakpoints for FLU were defined according to CLSI Document M27-A4 [43] and were as follows: susceptible, MIC ≤ 8 µg/mL; susceptible—dose-dependent, MIC = 16 to 32 µg/mL; and resistant, MIC ≥ 64 µg/mL. It was determined that 94.4% of *C. albicans* and all *C. krusei* strains were resistant to FLU. In contrast, 23.53% of the strains of C. glabrata were resistant, whereas all of the S. cerevisiae isolates were sensitive to FLU. Although CLSI and EUCAST have not established breakpoints for NY, several authors have proposed that topical NY MICs of less than 1 µg/mL can be regarded as susceptible [20,49]. If we apply this interpretation to our results, every strain chosen for this study was resistant to this antifungal agent.

The MIC values of MOEO for all the strains ranged between 0.156 mg/mL and 2.5 mg/mL. The most resistant were the *C. glabrata* strains: one had an MIC value of 2.5 mg/mL, and thirteen (76.74%) of them had an MIC value of 1.25 mg/mL. On the other hand, only one strain of *C. albicans* had an MIC value of 1.25 mg/mL, while the concentrations of MOEO, which inhibited the growth of most of the *C. albicans* strains, were 0.625 mg/mL (in 44.44%) and lower for the remaining half of the strains. One strain of *S. cerevisiae* had an MIC value of 0.312 mg/mL, and the other two had an MIC value of 0.156 mg/mL. In the case of the *C. krusei* strains, the minimum inhibitory concentrations were 0.625 mg/mL for 54.55% and 0.312 mg/mL for 45.45% of the total number of the tested strains.

### 3.3. Biofilm Production

The CV technique revealed that *C. albicans* strains had the highest capacity to produce biofilm of all the species analyzed (Table 2). Out of 66.67% of *C. albicans* strains that produced biofilm, 16.67% were strong, 22.22% moderate, and 27.78% weak biofilm producers. Among 52.94% of *C. glabrata* strains with biofilm production ability, 29.41% were moderate, and 23.53% were weak producers. Considering *S. cerevisiae* strains, only one was a weak producer, while none of the *C. krusei* strains could produce biofilm.

### 3.4. Antibiofilm Effects

The antibiofilm activities were tested for four *C. albicans* strains (Ca1, Ca2, Ca3, and Ca4; Nos. 1–4 from Table 2) and four *C. glabrata* (Cg1, Cg2, Cg3, and Cg4; Nos. 19–22 from Table 2) strains. Testing of the mature biofilm reduction via the two antimycotics and MOEO alone demonstrated that FLU had a very poor effect; many of the strains even exhibited increased biofilm production when treated with this agent (Figure 3). As one can notice, a concentration dependency was not observed, and the effects of various concentrations differed among the strains. The highest effect of this antimycotic was detected in the case of strain Ca3, followed by strain Ca1, while the lowest effect was observed for strains Ca2 and Cg1. All the *C. glabrata* strains and the Ca2 strain were unaffected by the FLU treatment, with statistical significance. Although effective for the two mentioned strains (Ca1 and Ca3) through reducing the amount of biofilm by as much as 43.39 and 66.76%, respectively, concentration dependency was noted only for strain Ca3.

In the case of NY, a much higher effect on mature biofilm reduction was detected, reducing up to 82.97% of the biofilm (strain Ca3). The efficiency of NY was the highest against strain Ca3, but similar values were noted for strains Ca1 and Ca4. On the other hand, the highest tolerance was observed for strains Cg1, Cg2, and Cg3. In only one strain (Cg3), the promotion of biofilm growth was detected, where the highest tested concentration increased biofilm formation by 20.21%. However, it should be emphasized that the two lower concentrations, MIC and MIC/2, effectively decreased the amount of biofilm by 39.46 and 40.19%, respectively. Concentration dependency was observed only for the *C. albicans* strains (Ca1, Ca2, Ca3, and Ca4), while this regularity did not exist in the case of the NAC strains (Figure 4).

MOEO showed the highest reduction, which was in the range of 18.24–95.36%, with the most prominent efficiency against strain Ca1. Promotion was observed for only one strain (Cg4), where 18.76% of the increase was measured for the MIC/2 concentration of MOEO. Concentration dependency was only noted for three strains, namely, Cg3, Ca2, and Ca4 (Figure 5).

### 3.5. Types of Interaction of the Melissa officinalis Essential Oil in Combination with Fluconazole or Nystatin

In order to investigate whether the observed MOEO antibiofilm effect could be utilized as an enhancer of conventional antifungal therapy, combined effects were studied between FLU and MOEO as well as between NY and MOEO (Figure 6). The results of the Bliss analysis showed strain-specific synergism in 37.5% of the strains for FLU and in 25% of the strains in the case of NY. The strains affected by combined MOEO and FLU demonstrated synergistic effects in 2 MIC, MIC, and MIC/2 concentrations of MOEO and in all concentrations of FLU (strains Ca2 and Cg4). On the other hand, in the case of strain Cg3, synergy was observed only for combinations of MIC/2 of MOEO and the lowest tested concentration of FLU. In the case of NY, the results only pointed to synergism for two strains (Ca2 and Cg4), where this type of interaction was observed for combinations of the two agents in concentrations of MIC and 2MIC (Ca2) and for MIC/4 and lower concentrations of FLU in combination with almost all concentrations of MOEO.

On the other hand, an antagonistic effect was also observed, and this was more prominent in the case of the FLU + MOEO combination but was also noted for the second combination. Even for the strains where synergistic interaction was found, some of the concentrations might cause the opposite effect. The highest antagonism was found in strains Ca1, Cg2, Cg3, and Ca4. In all these strains except Cg2, antagonism was found in combinations of MOEO at or below MIC/2 with all the tested concentrations of FLU. For strain Cg2, this effect was even more notable and stronger since it was found in all the combinations of the tested antimicrobials. NY, in combination with MOEO, also showed antagonism at lower concentrations of both agents, namely, below the MIC/2 of MOEO and the MIC/4 of NY.

## 4. Discussion

Vulvovaginal candidosis is one of the most common genital infections in women. In addition to the high prevalence of this mycosis, the incidence of recurrent forms of VVC is a significant concern. Considering the large number of female patients, most undergo treatment according to official recommendations, without undergoing mycological analyses. Multiresistant *Candida* strains are very important in the pathogenesis of RVVC because they are highly responsible for unsuccessful therapy. In the case of vulvovaginal isolates, one of the major concerns is resistance to the most commonly recommended and prescribed antimycotics, including FLU and NY.

Recently, in many studies that investigated the sensitivity of the most prevalent *Candida* species that cause VVC, high MIC values for FLU and NY were established. Choukri and colleagues reported that NY MICs for all species (*C. albicans*, *C. glabrata*, *C. krusei*, *C. tropicalis*, and *C. parapsilosis*) ranged as high as 4 µg/mL [50]. Similarly, Nelson et al. revealed that the most resistant species, i.e., *C. albicans*, *C. glabrata*, *and C. krusei*, showed sensitivity to FLU in a dose-dependent fashion (MIC = 64 µg/mL). Moreover, the MIC of NY needed to prevent the growth of the most resistant *C.albicans* and *C. glabrata* strains was 16 µg/mL, and this concentration for *C. krusei* was 4 µg/mL [49]. However, in other surveys, the results showed that NAC species, as well as the non-*Candida* yeast *S. cerevisiae*, necessitate the administration of high fungistatic concentrations of FLU (the MIC for *C. glabrata* and *C. krusei* was 128 µg/mL, and that for *S. cerevisiae* was 16 µg/mL). For the same isolates, the determined MICs of NY revealed resistance to this antimycotic (the MIC for *C. glabrata* was 16 µg/mL, that for C. *albicans* and *C. krusei* was 8 µg/mL, and that for *S. cerevisiae* was 4 µg/mL) [20].

In our study, the high MIC values of FLU and NY for the resistant vulvovaginal strains were up to 128 and 16 µg/mL, respectively. These findings demonstrated the emergence of azole- and poliene-resistant strains of yeasts, which are causative agents of recurrent VVC. The results of our survey and those of other studies on the same topic indicate the need to consider new antimycotics as well as natural products with antifungal properties. The majority of studies on the antimicrobial effects of MOEO have been conducted on bacteria, while there are fewer studies on the effects of this EO on *Candida* spp., none of which examined the effect on vulvovaginal yeast isolates. Raut et al. examined the effect of selected terpenoids on *C. albicans* ATCC 90028. Nine of these terpenoids were constituents of the EO used in this investigation. The minimum inhibitory concentration of Citral was 0.5 mg/mL; for both Citronellal and Myrcene, it was 1 mg/mL; for Linallol, α-Pinene, and β-Pinene, it was 2 mg/mL; for 1.8-Cineole, it was 4 mg/mL; and for both Geranyl acetate and p-Cymene, it was >4 mg/mL [39]. Dukić et al. determined the MIC of MOEO for one clinical *C. albicans* strain, equaling 0,3 mg/mL [34], and the same MIC value was obtained by Powers et al., who examined the effect of MOEO on a reference *C. albicans* (ATCC 18804) strain [51]. In the recent study conducted by Karpinski et al., the MIC values for MOEO were determined for different *Candida* spp. (*C. albicans*, *C. glabrata*, *C. krusei*, *C. parapsilosis*, *C. tropicalis*, and *C. guilliermondii*), and they ranged from 0.39 to 1.41 mg/mL [37]. In our study, the MIC values of MOEO for all strains were between 0.156 mg/mL and 2.5 mg/mL. The *C. glabrata* strains were the most resistant, with the highest MIC value amounting to 2.5 mg/mL, while the MIC for the majority of the strains was 1.25 mg/mL. The highest MIC values for the *C. albicans*, *C. krusei*, and *S. cerevisiae* strains were 1.25, 0.625, and 0.312 mg/mL, respectively.

In addition, it has been established that certain virulence factors, including the ability to avoid host defenses, adhesion ability, and the capacity for biofilm formation, contribute to the pathogenicity of *Candida* spp. [52]. The biofilm production of *Candida* spp. on the vaginal mucosa in vivo and its role in the VVC as well as in the recurrent form of this infection are still controversial topics. Although some reports have questioned this fact due to the lack of histological evidence [53,54], emphasizing that most of the experiments were conducted in vitro [25,55], numerous studies have suggested that the biofilm formation process, especially that of *C. albicans*, is very important with respect to the pathogenesis of VVC [16,18,19,56,57]. The study conducted by Wu et al. provided evidence to support this theory and demonstrated that the biofilm growth of *C. albicans* on the vaginal epithelium is associated with histological damage to mucosal epithelial cells and local inflammation [58]. Furthermore, there have been reports indicating that biofilms formed by *C. albicans* play a role in facilitating the development of persister cells, which are mostly responsible for resistance to antifungal medications, consequently leading to the development of RVVC [59,60].

The results of our study correlate with the findings of other authors examining the biofilm production of vulvovaginal *Candida* strains since the biofilm positivity for the *C. albicans* isolates reported herein is 66.67%. In comparison, only 32,23% of NAC isolates were found to be biofilm producers. Moreover, among the biofilm-positive isolates, the *C. albicans* strains had the highest biofilm production capacity because 16.67% and 22.22% were classified as strong and moderate producers, respectively, while none of the NAC species were strong, and 16.13% of them were moderate producers. Among the NAC species, *C. glabrata* strains accounted for 90% of the producers. Contrary to the fact that *C. krusei* is generally cited as a potent biofilm producer, in our study, none of the 11 tested vaginal isolates of this species had the ability to produce biofilms.

The ability of *Candida* species to form biofilms presents a great challenge for conventional antifungal therapy due to the existence of the extracellular matrix, changed gene expression (efflux pumps, ergosterol content, and changes in stress responses), and “persister” sessile cells. All of these factors contribute to the management of biofilm-associated *Candida* infections, eventually leading to frequent therapy failures [61]. Mature biofilms are especially important because they represent the most resistant form of microorganism communities and, at the same time, the site for further infection dispersal [62]. Therefore, the quest for new agents that might increase the effectiveness of the existing therapy, especially against mature forms of biofilms, which are highly expected to develop in chronic forms of infections, is very important. In the present research, the effect of the drugs FLU and NY were assayed for their efficiency against various *Candida* spp. mature biofilms and compared to the effect of MOEO alone against the same biofilms. The results revealed the much more proficient action of the MOEO, which reduced up to 95.36% of the biofilm. However, it has to be mentioned that NY also exhibited very good antibiofilm action against all strains, reducing their biofilm mass by as much as 82.97%.

In the case of FLU, the effect was much lower, and in many cases, biofilm growth was promoted instead of reduced. This phenomenon is related to increased gene expression under stress conditions, resulting in greater action of the efflux pumps and triggering other survival mechanisms [63]. Also, this should be connected with the high resistance of these strains to FLU, which is usually even increased under biofilm conditions, as mentioned above. Due to the fact that relatively low concentrations of FLU were used for antibiofilm testing in such resistant strains, other results could not be expected. Another important conclusion of the herein-obtained results is that the albicans strains showed significantly higher reduction by all agents compared to the NAC strains. This is a consequence of their higher biofilm production capability (Table 2), which was up to 17 times higher than that of the NAC strains. Previous investigations on the subject of *Candida* biofilm eradication via FLU have been conducted in several studies, where the authors reported antibiofilm concentrations in the range of from 4 to ≥1024 µg/mL [39,64,65,66,67,68]. In our study, the MICs for FLU were not determined since the reduction did not exceed 80%, as required for the determination of this concentration in the case of biofilms [66].

In contrast to FLU, far fewer studies have examined the effects of NY on mature *Candida* biofilms, and it was determined that this antimycotic was effective at concentrations ranging from 4 to 128 µg/mL [69,70]. In our study, a biofilm reduction level of 80%, as required for the MIC determination of biofilms according to Li et al., was observed in three isolates, Ca1, Ca3, and Ca4, with MICs of 4, 8, and 16 g/mL, respectively [66]. The differences between the antibiofilm efficacy of FLU and NY might result from the distinction in their mechanisms of action. While FLU targets the de novo synthesis of ergosterol by affecting the enzymes of this process, NY binds the ergosterol already present in the fungal cells and consequently damages already-formed biofilm [71].

Although there are no studies about the effect of MOEO on mature *Candida* biofilms in the literature, the already-mentioned study by Raut et al. examined the effect of selected terpenoids, besides planktonic cells, on *C. albicans* ATCC 90028 mature biofilm. The elimination of the biofilm network caused by the terpenoid treatment was indicated by an absorbance reduction of >50% compared to that of the control. The minimum inhibitory concentration of Citral and Linallol was 2 mg/mL, that of Citronellal was 4 mg/mL, and that for 1.8-Cineole, Geranyl acetate, α-Pinene, β-Pinene, p-Cymene, and Myrcene was > 4 mg/mL [37]. In our study, the MICs for strain Ca1′s and Ca3′s biofilms were 1.25 and 0.312 mg/mL, respectively. For the other two *C. albicans* strains, the MICs were >1.25 mg/mL, and for all the *C. glabrata* strains, the MIC was > 2.5 mg/mL, which means that MOEO had a stronger effect on the *C. albicans* biofilms compared to those formed by *C. glabrata*.

The increasing resistance of *Candida* strains to antifungal drugs represents a major challenge for therapy for fungal infections, especially chronic ones, which are related to biofilm production.

Therefore, enhancing the effectiveness of one antimicrobial by combining its action with another antimicrobial will provide a higher potential for antifungal therapy. Moreover, if this potentiator can reduce the growth of the strains resistant to one agent, the significance of this affordance is even greater. As each of the agents tested in the present study (MOEO, NY, and FLU) induced good or slight reductions (FLU) in the mature *Candida* biofilms, the investigation of the combined action of these drugs and MOEO against mature biofilms was performed. In this assay, the goal was to find combinations that might enhance the action of FLU or NY and increase the success of therapy using these two drugs, especially for enhancing their efficacy against mature biofilms of multi-resistant strains. The results showed strain-related effects and high concentration dependency since, in the same strains, the effect of the same two compounds could be both synergistic as well as antagonistic. Especially notable efficacy was noted in the case of strain Ca2, where FLU induced a reduction in mature biofilm when combined with MOEO at MIC and 2MIC concentrations, but alone, it had a promotive effect, as visible in Figure 6. Also, it must be emphasized that in the analysis reported here, we calculated all the combinations of concentrations, allowing for a much deeper understanding of the interactions that take place. Up to now, synergism against candidal biofilms has been tested for certain natural compounds, such as pseudolaric acid A [72]; catechins [73]; paeonol [68]; allyl isothiocyanate [74]; thymol and eugenol [75]; carvacrol, thymol, and eugenol [76]; allicin [77]; and other natural compounds of plant origin such as quercetin [29] or shikonin [78]. Up to now, the only study that investigated the combined effect of essential oils and antifungal drugs on candidal strains was performed by Jafri and Ahmad, where synergism was found between *Thymus vulgaris* and thymol against drug-resistant strains of *C. albicans* and *C. tropicalis* [79]. However, in most of these studies, the authors performed classical calculations of the fractional inhibitory concentration index (FICI), which does not provide further information about effects among lower doses, which might highly influence therapeutic success.

It has been reported that terpenoids (major compounds and the main agents of antimicrobial action) can sensitize *Candida* biofilms to FLU, where among thymol, menthol, and eugenol, the first one demonstrated the highest synergy in reducing biofilm formation, with an FICI of 0.37 [75]. Another study on terpenoid action against candidal biofilms determined that there was synergy between eugenol and FLU as well as between carvacrol and FLU [76]. However, the endpoint for MIC determination was 50% of the control mature biofilm, and the FICI values were 0.374 for carvacrol and FLU and 0.312 for eugenol and FLU. The same criteria were applied in a study where allyl isothiocyanate was tested for its synergy with FLU and in which the FICI values ranged from 0.132 to 0.312 84 [74]. Thymol and the essential oil of *Thymus vulgaris* were found to be synergistic in half-MIC concentrations and significantly reduced biofilm formation in candida strains [79]. Asiatic acid, a compound found in many plants, was also tested for its synergy with FLU against FLU-resistant *C. albicans*, and an FICI of 0.25 (in reference to an MIC defined as an 80% reduction in the control biofilms) was reported [80]. However, this study was investigating the development of biofilms and not mature ones, which present much more resistant forms than developing biofilms. The main conclusion that might be drawn is that MOEO has high potential in reducing the mature biofilms of multi-drug-resistant *Candida* strains, even against those whose biofilms demonstrated resistance to FLU alone (strains Cg4 and Ca2) and NY (strain Cg4). Also, the previous studies focused on only strain one or fewer, while here, a total of eight strains were tested for synergy, revealing the strain-specific action of the investigated combinations. Therefore, the potential design of supplemented or combined therapy should be performed with caution since antagonism has been observed for certain concentrations of the combined substances that showed synergy.

## 5. Conclusions

In the present study, the antifungal and antibiofilm effects of MOEO, FLU, and NYS alone or in combination against strains that are causative agents of RVVC were investigated for the first time. The isolates that showed high percentages of resistance to FLU and all those that demonstrated resistance to NY were susceptible to MOEO in concentrations ranging from 0.156 to 2.5 mg/mL. The testing of their effects on mature biofilm revealed very limited action of FLU, which even demonstrated promotive action with respect to the metabolic activity of biofilm cells in certain strains, while NY and MOEO reduced such cells’ activity by as much as 17.7% and 4.6%, respectively. Finally, the testing of the MOEO and drugs’ combined effect showed strain-specific synergistic action, and in many cases, the lower concentrations demonstrated antagonistic effects, even in those strains where synergism was detected. Therefore, the potential design of supplemented or combined therapy should be performed cautiously, especially at the applied concentrations, to avoid possible antagonism and therapeutic failure.

## Figures and Tables

**Figure 1 jof-09-01080-f001:**
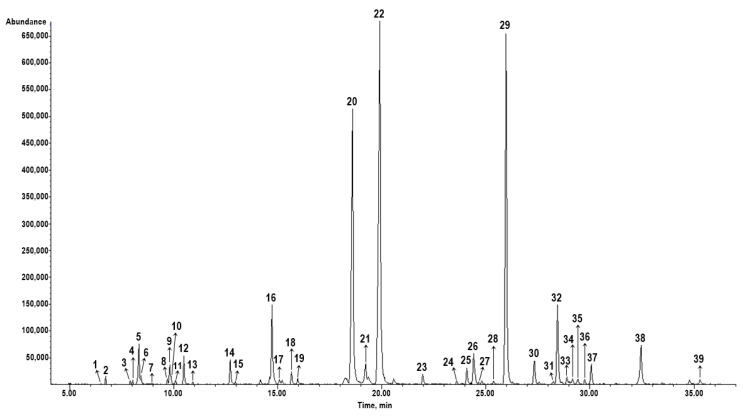
GC/MS chromatogram of *Melissa officinalis* essential oil.

**Figure 2 jof-09-01080-f002:**
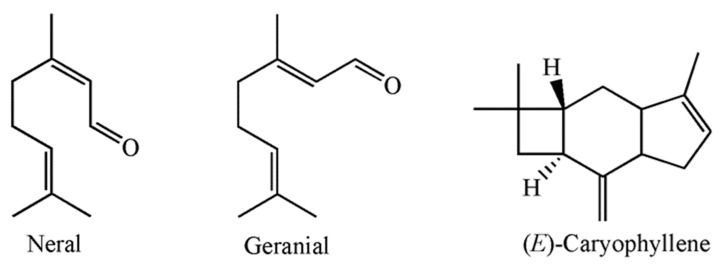
Chemical structures of the most abundant components in *Melissa officinalis* essential oil.

**Figure 3 jof-09-01080-f003:**
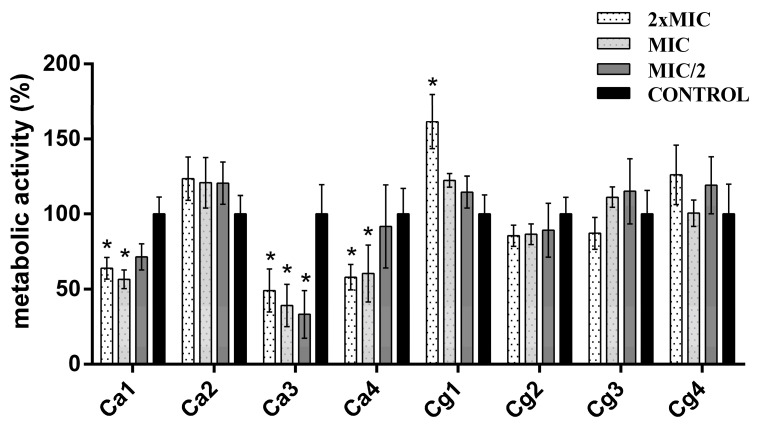
Antibiofilm effect of fluconazole. Ca1, Ca2, Ca3, and Ca4, *C. albicans* strains; Cg1, Cg2, Cg3, and Cg4, *C. glabrata* strains. * Statistical significance (*p <* 0.05).

**Figure 4 jof-09-01080-f004:**
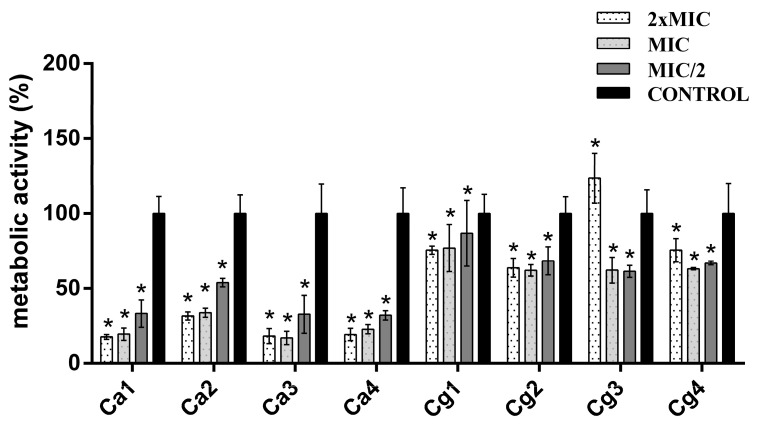
Antibiofilm effect of nystatin. Ca1, Ca2, Ca3, and Ca4, *C. albicans* strains; Cg1, Cg2, Cg3, and Cg4, *C. glabrata* strains. * Statistical significance (*p <* 0.05).

**Figure 5 jof-09-01080-f005:**
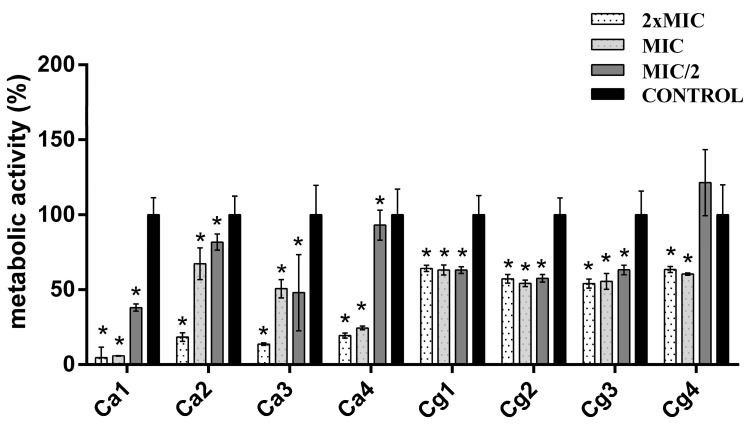
Antibiofilm effect of *Melissa officinalis* essential oil. Ca1, Ca2, Ca3, and Ca4, *C. albicans* strains; Cg1, Cg2, Cg3, and Cg4, *C. glabrata* strains. * Statistical significance (*p <* 0.05).

**Figure 6 jof-09-01080-f006:**
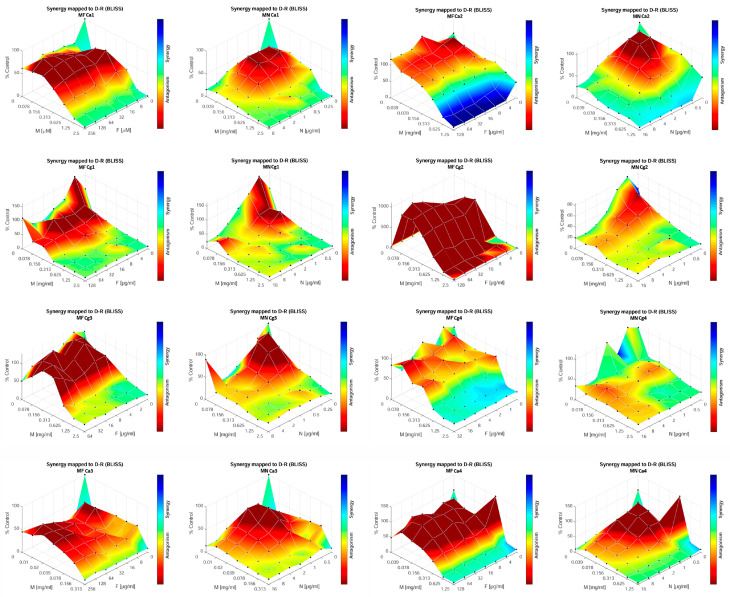
Types of interaction of the *Melissa officinalis* essential oil in combination with fluconazole or nystatin. F, fluconazole; N, nystatin; M, *Melissa officinalis* essential oil; Ca1, Ca2, Ca3, and Ca4, *C. albicans* strains; Cg1, Cg2, Cg3, and Cg4, *C. glabrata* strains.

**Table 1 jof-09-01080-t001:** Chemical composition of *Melissa officinalis* essential oil.

No.	t_ret_, min	Compound	RI^exp^	RI^lit^	Method of Identification	Content, %
**1**	6.50	α-Thujene	926	924	RI, MS	tr
**2**	6.72	α-Pinene	934	932	RI, MS, Co-I	0.2
**3**	7.90	Sabinene	974	969	RI, MS	tr
**4**	8.03	β-Pinene	979	974	RI, MS, Co-I	tr
**5**	8.32	6-Methyl-5-hepten-2-one	989	981	RI, MS	2.2
**6**	8.42	Myrcene	993	988	RI, MS	tr
**7**	8.95	(3*E*)-Hexenyl acetate	1008	1001	RI, MS	tr
**8**	9.69	*p*-Cymene	1028	1020	RI, MS	tr
**9**	9.80	Limonene	1028	1024	RI, MS, Co-I	0.9
**10**	9.85	1,8-Cineole	1033	1026	RI, MS, Co-I	tr
**11**	10.08	(*Z*)-β-Ocimene	1039	1032	RI, MS	tr
**12**	10.48	(*E*)-β-Ocimene	1049	1044	RI, MS	1.2
**13**	10.90	γ-Terpinene	1061	1054	RI, MS, Co-I	tr
**14**	12.71	Linalool	1105	1095	RI, MS, Co-I	1.3
**15**	12.96	*cis*-Rose oxide	1114	1106	RI, MS	tr
**16**	14.71	Citronellal	1156	1148	RI, MS	4.6
**17**	15.08	*trans*-Pinocamphone	1165	1158	RI, MS	tr
**18**	15.64	*cis*-Pinocamphone	1179	1172	RI, MS	0.6
**19**	15.96	(*E*)-Isocitral	1186	1177	RI, MS	0.4
**20**	18.59	Neral (*Z*-Citral)	1245	1235	RI, MS, Co-I	19.7
**21**	19.21	Methyl citronellate	1263	1257	RI, MS	1.1
**22**	19.91	Geranial (*E*-Citral)	1274	1264	RI, MS, Co-I	31.0
**23**	21.96	Methyl geranate	1327	1322	RI, MS	0.7
**24**	23.59	Neryl acetate	1367	1359	RI, MS	tr
**25**	24.07	α-Copaene	1378	1374	RI, MS	1.2
**26**	24.41	Geranyl acetate	1386	1379	RI, MS	2.0
**27**	24.65	β-Cubebene	1392	1387	RI, MS	tr
**28**	25.36	(*Z*)-Caryophyllene	1409	1408	RI, MS	tr
**29**	25.98	(*E*)-Caryophyllene	1425	1417	RI, MS	19.4
**30**	27.32	α-Humulene	1458	1452	RI, MS	1.4
**31**	28.23	γ-Muurolene	1481	1478	RI, MS	tr
**32**	28.45	Germacrene D	1486	1487	RI, MS	5.3
**33**	28.88	Bicyclogermacrene	1497	1500	RI, MS	0.6
**34**	29.16	α-Muurolene	1504	1500	RI, MS	tr
**35**	29.41	(*E*,*E*)-α-Farnesene	1510	1505	RI, MS	0.4
**36**	29.75	γ-Cadinene	1519	1513	RI, MS	0.3
**37**	30.05	δ-Cadinene	1527	1522	RI, MS	1.3
**38**	32.45	Caryophyllene oxide	1590	1582	RI, MS	3.5
**39**	35.29	α-Cadinol	1662	1652	RI, MS	0.5
					Total identified (%)	99.8
	Grouped components (%)					
	Monoterpene hydrocarbons (1–4, 6, 8, 9, 11–13)				2.3	
	Oxygen-containing monoterpenes (10, 14–24, 26)				61.4	
	Sesquiterpene hydrocarbons (25, 27–37)				29.9	
	Oxygen-containing sesquiterpenes (38, 39)				4.0	
	Others (5, 7)				2.2	

t_ret_.: retention time; RI^lit^: retention indices from the literature (Adams, 2007 [42]); RI^exp^: experimentally determined retention indices on HP-5MS column by co-injection of a homologous series of n-alkanes C_8_–C_20_. MS: constituent identified by mass-spectra comparison; RI: constituent identified by retention index matching; Co-I: constituent identity confirmed by GC co-injection of an authentic sample; tr = trace amount (<0.05%).

**Table 2 jof-09-01080-t002:** Broth microdilution assay and categorization of biofilm production.

Isolate No.	*Candida* Species	Biofilm Producer Category	NY MIC (µg/mL)	FLU MIC (µg/mL)	MOEO MIC (mg/mL)
**1**	*C. albicans*	strong	4	128	1.25
**2**	*C. albicans*	strong	8	64	0.625
**3**	*C. albicans*	strong	8	128	0.156
**4**	*C. albicans*	moderate	8	64	0.625
**5**	*C. albicans*	moderate	8	128	0.156
**6**	*C. albicans*	moderate	8	128	0.156
**7**	*C. albicans*	moderate	8	128	0.312
**8**	*C. albicans*	weak	8	32	0.312
**9**	*C. albicans*	weak	8	128	0.625
**10**	*C. albicans*	weak	2	128	0.156
**11**	*C. albicans*	weak	4	128	0.312
**12**	*C. albicans*	weak	16	64	0.625
**13**	*C. albicans*	no	8	128	0.625
**14**	*C. albicans*	no	8	128	0.625
**15**	*C. albicans*	no	8	128	0.625
**16**	*C. albicans*	no	8	128	0.312
**17**	*C. albicans*	no	4	128	0.312
**18**	*C. albicans*	no	8	128	0.625
**19**	*C. glabrata*	moderate	8	64	1.25
**20**	*C. glabrata*	moderate	8	64	1.25
**21**	*C. glabrata*	moderate	4	32	1.25
**22**	*C. glabrata*	moderate	8	16	1.25
**23**	*C. glabrata*	moderate	4	16	1.25
**24**	*C. glabrata*	weak	4	16	1.25
**25**	*C. glabrata*	weak	4	32	1.25
**26**	*C. glabrata*	weak	4	64	1.25
**27**	*C. glabrata*	weak	8	64	0.312
**28**	*C. glabrata*	no	4	4	1.25
**29**	*C. glabrata*	no	8	4	1.25
**30**	*C. glabrata*	no	4	8	1.25
**31**	*C. glabrata*	no	4	4	1.25
**32**	*C. glabrata*	no	8	2	2.5
**33**	*C. glabrata*	no	8	8	1.25
**34**	*C. glabrata*	no	8	32	0.625
**35**	*C. glabrata*	no	8	8	0.625
**36**	*S. cerevisiae*	weak	8	4	0.156
**37**	*S. cerevisiae*	no	4	8	0.312
**38**	*S. cerevisiae*	no	4	2	0.156
**39**	*C. krusei*	no	8	64	0.625
**40**	*C. krusei*	no	8	64	0.625
**41**	*C. krusei*	no	4	128	0.625
**42**	*C. krusei*	no	4	64	0.312
**43**	*C. krusei*	no	8	64	0.312
**44**	*C. krusei*	no	8	64	0.312
**45**	*C. krusei*	no	16	128	0.625
**46**	*C. krusei*	no	8	64	0.312
**47**	*C. krusei*	no	8	64	0.625
**48**	*C. krusei*	no	4	64	0.312
**49**	*C. krusei*	no	8	64	0.625

## Data Availability

The data presented in the study are available on request from the corresponding author.

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
