# Peer review of "Antifungal Activity and Type of Interaction of Melissa officinalis Essential Oil with Antimycotics against Biofilms of Multidrug-Resistant Candida Isolates from Vulvovaginal Mucosa"

_jof, 2023, doi:10.3390/jof9111080_

Round 1

Reviewer 1 Report

The present manuscript focuses on the anti-biofilm activity of Melissa officinalis essential oil (MOEO) against multidrug-resistant Candida isolates from vulvovaginal mucosa, both alone and in combination with the conventional antimycotics fluconazole and nystatin. The authors found that MOEO alone displayed a very good anti-biofilm activity. However, when combined with fluconazole or nystatin, MOEO showed strain-specific synergistic action, and in many cases, the lower concentrations demonstrated antagonistic effects, even in those strains where synergism has been detected, suggesting that potential combined therapeutic strategies should be done cautiously to avoid therapy failure.

The topic is very interesting and suitable for the Special Issue” Treatment of Oral and Vulvovaginal Candidiasis by Natural Product”. However, in my opinion, some points need to be addressed before publication.

Major comments:

1.      Title “multi-resistant Candida isolates” I would suggest “ multidrug-resistant”

2.      Abstract: line 27 ” while all strains showed resistance to fluconazole” maybe “ while all strains showed resistance to nystatin?”

3.      Material and methods section: lines 160-161 “ For each strain, the microdilution test was performed three times” what do the authors mean? in triplicates for each experimental point or three independent experiments? Please clarify and add statistical analysis in Table 2.

4.      Material and Methods. Section 2.5. I would suggest to summarise the three paragraphs into two paragraphs, including the methods used for the biofilm evaluation  along with treatments, as follows:

2.5.1 quantification of biofilm biomass by CV staining

2.5.2.  evaluation of biofilm metabolic activity by XTT assay

3.  Can the authors clarify if appropriate vehicle controls (DMSO) were used for MIC and biofilm inhibition assays?

4.  The data reported in Figures 3,4 and 5 are the means of how many experiments? Please, specify and provide statistical analysis.

5.  Figure 6 is unreadable.  Please, improve the resolution and quality.

6.  Please update the reference n. 42. Please see “Alexander B.D. Reference Method for Broth Dilution Antifungal Susceptibility Testing of Yeasts. 4th ed. Clinical and Laboratory Standards Institute; Wayne, PA, USA: 2017. Clinical and Laboratory Standards Institute (CLSI) standard M27”.

7. Among natural compounds displaying antibiofilm activity against C. albicans (lines 79-80), retinoids should be also included. Please see the following article “Pistoia, E. et al. (2022). All-Trans Retinoic Acid Effect on Candida albicans Growth and Biofilm Formation. Journal of Fungi, 8(10), 1049. doi.org/10.3390/jof8101049

9.  Software version in the material and method section is missing

Minor comments

“ml” please “mL”

Lines 242-244 should be removed.

Minor grammatical and font errors are presented in the main text. Please check the Lines 98, 268, 273, 274. “Candida spp” should be italicized.

Minor editing of English language is required

Author Response

Dear reviewer,

Thank you very much for your consideration that our topic is very interesting and suitable for the Special Issue” Treatment of Oral and Vulvovaginal Candidiasis by Natural Product.” Additionally, thank you for your purposeful suggestions, which we applied to improving the paper. All changes that follow your suggestion are marked green.As for major comments:

  1. Title “multi-resistant Candidaisolates” I would suggest “ multidrug-resistant”

Response: We changed now “multi-resistant Candida isolates” into “multidrug-resistant”

  1. Abstract: line 27 ” while all strains showed resistance to fluconazole” maybe “ while all strains showed resistance to nystatin?”

Response: Th Thank you for this suggestion. It was a technical mistake, and we changed it now.

  1. Material and methods section: lines 160-161 “ For each strain, the microdilution test was performed three times” what do the authors mean? in triplicates for each experimental point or three independent experiments? Please clarify and add statistical analysis in Table 2.

Response: It is changed now into triplicates. Statistical analysis was not performed due to no differences between the replicates.

  1. Material and Methods. Section 2.5. I would suggest to summarise the three paragraphs into two paragraphs, including the methods used for the biofilm evaluation  along with treatments, as follows:

2.5.1 quantification of biofilm biomass by CV staining

2.5.2.  evaluation of biofilm metabolic activity by XTT assay

Response: It is changed now

  1. Can the authors clarify if appropriate vehicle controls (DMSO) were used for MIC and biofilm inhibition assays?

Response: Thank you very much for this suggestion. Yes, we had appropriate vehicle controls (DMSO) that were used for MIC and biofilm inhibition assays. It is now changed into “In this assay, vehicle control with only the solvent (DMSO) was performed to exclude the possible effect of solvent activity.”

  1. The data reported in Figures 3,4 and 5 are the means of how many experiments? Please, specify and provide statistical analysis.

Response: The data are the results of the triplicate experiments, and statistical analysis is provided now, and the methods of statistical analysis are provided in section 2.7.

  1. Figure 6 is unreadable.  Please, improve the resolution and quality.

Response: The resolution of the Figure 6 is set to maximum quality now.

  1. Please update the reference n. 42. Please see “Alexander B.D. Reference Method for Broth Dilution Antifungal Susceptibility Testing of Yeasts. 4th ed. Clinical and Laboratory Standards Institute; Wayne, PA, USA: 2017. Clinical and Laboratory Standards Institute (CLSI) standard M27”.

Response: It is changed now

  1. Among natural compounds displaying antibiofilm activity against C. albicans (lines 79-80), retinoids should be also included. Please see the following article “Pistoia, E. et al. (2022). All-Trans Retinoic Acid Effect on Candida albicans Growth and Biofilm Formation. Journal of Fungi, 8(10), 1049. doi.org/10.3390/jof8101049

Response: It is included now in the reference list.

  1. Software version in the material and method section is missing

Response: It is included now

Minor comments

“ml” please “mL”

Response: It is changed now

Lines 242-244 should be removed.

Response: It is removed now

Minor grammatical and font errors are presented in the main text. Please check the Lines 98, 268, 273, 274. “Candida spp” should be italicized.

 Response: It is changed now

The corrected manuscript is in the attachment.

Reviewer 2 Report

General Impressions

The authors describe the results of a study on the effects of a preparation of essential oil from Melissa officinalis (MOEO) on growth and biofilm formation of Candida species. While the effects of MOEO on Candida sp. have been described previously, the current study expands on the existing body of knowledge by examining the synergistic effects between MOEO and two antifungal drugs, fluconazole and nystatin. The synergy study has been well designed and execute, but the findings are ultimately not clear. The authors describe that, depending on the strain and the drug, the MOEO drug combinations combination can show antagonism, no effect or synergism. In this, the study does not add much to the existing body of knowledge and has very limited value for the scientific community.

Major points

1.      Description of Synergy: The authors dedicate a large part of the manuscript to the description and discussion of synergies, which this reviewer found difficult to follow. A table would be more more useful than the text or Figure 6 to summarize the complex and contradictory findings shown in figure 6.

2.      FICI scores vs. Bliss method: The authors express a preference for the Bliss method as it is described as more sensitive – that is, showing interactions that are not evident in FICI scoring. However, the interactions presented here do not show a trend or common principles. This begs the question if the chosen method might have been too sensitive – that is, showing interactions where they do not exist (type 1 errors). A discussion of FICI scores of the MOEO-FZ/NY interactions would be helpful – it might show that some of the Bliss findings are not significant and do not need to be explained.

3.      Biofilm vs. adhesion: The authors conclude that MOEO treatment reduces biofilm formation by Candida species. However, the method does depend on the detection of microplate-adherent cells. The addition of oils can be expected to reduce adhesion to surfaces, and the method thus does not allow to rule out that biofilms form but do not adhere to microtiter plates. Can a microscopic image be included to show the morphology of MOEO-treated cells? Does the oil prevent hypha-formation?

4.      Significance (Figs 3-5). The drug effects are not quite clear in some strains and concentration ranges. Please include data on the significance of the findings. The big picture might become clearer once the discussion is limited to significant findings.

Other points

1.      Writing style: The experimental data show no clear picture of MOEO/drug synergies, and the authors engage in a lengthy, speculative discussion about the results. The resulting discussion section is far too long.

2.      Please check the references. I tried to look up the description of the Biofilm method, which is cited as “Stepanovic et al [3]”. In the reference section, citation [3] is not Stepanovic et al. Something went wrong here and makes me wonder if the rest of the references are accurate.

3.      Style, grammar and layout need improvement. E.g., the template text for “Results” was not removed, lines 26-28 contrast drug resistance findings by comparing fluconazole to fluconazole.

4.      Toxicity of MOEO: Do we know the effects of MOEO on cell survival? That would help explain the findings (dead cells do not form biofilms). Please discuss MOEO toxicity.

Language, grammar and organization of text need improvement (see review comments).

Author Response

Dear reviewer,

Your suggestions for improving our paper are very helpful and constructive. Thank you very much for your revision. We changed the disadvantages following your suggestions and comments. All changes are in the attached document.

As for major points

  1. and 2. Description of Synergy: The authors dedicate a large part of the manuscript to the description and discussion of synergies, which this reviewer found difficult to follow. A table would be more more useful than the text or Figure 6 to summarize the complex and contradictory findings shown in figure 6. FICI scores vs. Bliss method: The authors express a preference for the Bliss method as it is described as more sensitive – that is, showing interactions that are not evident in FICI scoring. However, the interactions presented here do not show a trend or common principles. This begs the question if the chosen method might have been too sensitive – that is, showing interactions where they do not exist (type 1 errors). A discussion of FICI scores of the MOEO-FZ/NY interactions would be helpful – it might show that some of the Bliss findings are not significant and do not need to be explained.
  2.  

Response: Thank you very much for your suggestion. We absolutely agree that our presentation of results and comparison of FICI scores vs. Bliss method could induce confusion. For better understanding, we presented the modeling methods by which we performed the analysis. In spate, we performed FICI scoring for each combination. We decided to include modeling, which would give extensive results, mainly because it gives results that cannot be read using the FICI scoring. In order not to confuse the readers, we have decided to exclude this part of the discussion.

  1. Biofilm vs. adhesion: The authors conclude that MOEO treatment reduces biofilm formation by Candida species. However, the method does depend on the detection of microplate-adherent cells. The addition of oils can be expected to reduce adhesion to surfaces, and the method thus does not allow to rule out that biofilms form but do not adhere to microtiter plates. Can a microscopic image be included to show the morphology of MOEO-treated cells? Does the oil prevent hypha-formation?

 Response: We do not conclude that MOEO reduces biofilm formation. We have tested the effect of MOEO on preformed, mature biofilms. Therefore, the potential of MOEO to decrease the adhesion was not tested. Microscopic imaging was not performed, and it might be a subject of our future studies.

  1. Significance (Figs 3-5). The drug effects are not quite clear in some strains and concentration ranges. Please include data on the significance of the findings. The big picture might become clearer once the discussion is limited to significant findings.

 Response: We have now included statistical analysis results in figures 3-5 and changed the text where non-significant results were discussed.

Other points

  1. Writing style: The experimental data show no clear picture of MOEO/drug synergies, and the authors engage in a lengthy, speculative discussion about the results. The resulting discussion section is far too long.

 Response: The discussion is shortened now, and the speculations are excluded.

  1. Please check the references. I tried to look up the description of the Biofilm method, which is cited as “Stepanovic et al [3]”. In the reference section, citation [3] is not Stepanovic et al. Something went wrong here and makes me wonder if the rest of the references are accurate.

 Response: Something went wrong with the citation program, so it is fixed now, and this reference is No 46. Two references about the used method are also added (44 and 45). Thank you for the suggestion.

  1. Style, grammar and layout need improvement. E.g., the template text for “Results” was not removed, lines 26-28 contrast drug resistance findings by comparing fluconazole to fluconazole.

Response: It is removed now

  1. Toxicity of MOEO: Do we know the effects of MOEO on cell survival? That would help explain the findings (dead cells do not form biofilms). Please discuss MOEO toxicity.

Response: Due to the large number of the tested strains, our investigation did not include MFC determination, but we performed an MTT test that detected only viabile cells in the mature biofilms.

Reviewer 3 Report

The manuscript by Randelovic et al, is clearly and carefully written, materials and methods contain the required detail to reproduce the work, results are clearly presented and described. The experimental work appears to have been carried out well, with the exception of the biofilm quantification by crystal violet assay and the number of independent experiments seems to be only one (although performed in triplicates). I suggest the following revisions:

Line 26: “Antimicrobial testing showed that 94.4% of Candida albicans and all Candida krusei isolates were resistant to fluconazole, while all strains showed resistance to 27 fluconazole.” Please, consider revising.

Line 55: “Saccharomyces (S.) cerevisiae”: (S.)?

Line 98: Melissa officinalis: must be in italic

Line 101: Melissa officinalis in an italic title should not be in italic

Line 159: 0.5–2.5 × 105 CFU/mL -> 0.5–2.5 × 105 CFU/mL

Line 161: the standard recommend incubation at 35ºC and not 37ºC. If the authors did not follow closely the M27-A2 standards, then in line 149, it should be stated that “was tested by using broth microdilution method [42]” with slight modifications.

Line 173: Biofilm producing ability: In this assay, which published protocol did the author follow? Previously to the addition of CV, did the author fixed the biofilm with methanol? Line 182, after removing the excess of dye, did the authors wash 3 times with PBS (this step is crucial for a correct quantification)? 

Line 182: “250 μl of 96% (v/v) ethanol was added to each well for 45 min. After destaining, 150 μl of the prepared solutions was transferred into a sterile microtiter plate.” The biofilms did not dissolve totally in acetic acid 33% after 20 min of incubation to be read in the same microtiter plate? In general, they do, but effectively sometimes, they don’t because the biomass is too high. Please explain the alteration that were introduced to this protocol.

It is important to mention that there is no destaining in this protocol, instead, we aim to quantify all the staining. Higher the staining, higher the biofilm.

Line 179: “105” -> 10cells/mL?

Line 242-244: to be removed 

Fig 1, Fig 2 and Table 1 legend: “lemon balm essential oil”? or Melissa officinalis essential oil?

Line 272- 360: From line 272 until line 360, none of the fungi name or even Melissa officinalis are in italic as they should.

Figure 3,4,5: Y axis: the assays performed did not quantify a growth %, but the removal of a biofilm, measured by the remaining biofilm metabolic activity, expressed in % in relation to control.

The increased of the metabolic activity of the biofilms depending on the concentration of the antifungal compound or EO are difficult to understand. Did this happen in each replicate? The material and methods mentioned triplicates, but did the authors performed triplicates of 3 independent assays? If so, why did not they represent error bars?

Paragraph line 333: The synergy analysis was interpreted by Bliss analysis. The justification for this approach is explained in the discussion and an example of a correlation with the FICI determination is presented in the discussion. Could the authors include in supplementary results a table with the FICI of all the combinations used? So a reader could compare more easily with the literature? Moreover, not all the readers are able to understand the Figure 6. Moreover, the image quality must be increased. In this pdf version, I was not able to read the legends of each combination.

Line 361: Discussion

The discussion is very large, for example, line 384, there is no need to repeat the results. Although, the discussion is large, it is also very well written, present a good interpretation of the results and answers to questions that the readers may had along the paper.

Line 447: I agree with the discussion but we also have to be aware that the effect of antifungals in the biofilms also depends on their mechanism of action. The nystatin exhibited a good antibiofilm activity because it binds the ergosterol already present in the fungal cells of the biofilm, leading to the fungal death. Yet, the fluconazole target enzymes of the synthesis of ergosterol. So it targets the de novo synthesis of ergosterol, with no effect on the cells already present in the biofilm. 

Line 517: I do not understand how the FICI calculation do not provide further information about effects at lower concentrations. No reference is cited to support this. For a reader, FICI calculation can be run with all the concentrations tested in the checkerboard assay as well. 

Comment: The authors presented the Fig 3, 4 and 5 instead of determination of a value of SMIC because it was not possible to eradicate all the biofilm at the concentrations tested? Why did not they increase the concentration above 2x MIC?

Author Response

Dear reviewer,

Thank you for your consideration that our manuscript is clearly and carefully written, materials and methods contain the required detail to reproduce the work, and results are clearly presented and described. The corrected manuscript is in the attachment. All changes that follow your suggestion are marked yellow.

Line 26: “Antimicrobial testing showed that 94.4% of Candida albicans and all Candida krusei isolates were resistant to fluconazole, while all strains showed resistance to 27 fluconazole.” Please, consider revising.

Response: It is changed now. Thank you for the suggestion.

Line 55: “Saccharomyces (S.) cerevisiae”: (S.)?

Response: It is changed now to “Saccharomyces cerevisiae (S. cerevisiae)”

Line 98: Melissa officinalis: must be in italic

Response: It is changed now

Line 101: Melissa officinalis in an italic title should not be in italic

Response: It is changed now

Line 159: 0.5–2.5 × 105 CFU/mL -> 0.5–2.5 × 105 CFU/mL

Response: It is changed now

Line 161: the standard recommend incubation at 35ºC and not 37ºC. If the authors did not follow closely the M27-A2 standards, then in line 149, it should be stated that “was tested by using broth microdilution method [42]” with slight modifications.

Response: It is changed now

Line 173: Biofilm producing ability: In this assay, which published protocol did the author follow? Previously to the addition of CV, did the author fixed the biofilm with methanol? Line 182, after removing the excess of dye, did the authors wash 3 times with PBS (this step is crucial for a correct quantification)? 

Response: We did not fix the cells with methanol. The protocol that we followed was the one that was repeated many times in the papers regarding the effects of essential oils on fungal cells. We added a reference regarding this protocol. The protocol includes washing three times, and it was mentioned in line 203.

Line 182: “250 μl of 96% (v/v) ethanol was added to each well for 45 min. After destaining, 150 μl of the prepared solutions was transferred into a sterile microtiter plate.” The biofilms did not dissolve totally in acetic acid 33% after 20 min of incubation to be read in the same microtiter plate? In general, they do, but effectively sometimes, they don’t because the biomass is too high. Please explain the alteration that were introduced to this protocol.

It is important to mention that there is no destaining in this protocol, instead, we aim to quantify all the staining. Higher the staining, higher the biofilm.

Response: We did not apply acetic acid because we followed the protocol mentioned in the previous response. Considering destaining, we replaced this term in line 184: “After removing excess dye, the remaining CV in biofilms was extracted by adding 250 µl of 96% (v/v) ethanol to each well for 45 min. After this step, 150 µl of the prepared solutions were transferred into a sterile microtiter plate.”

Line 179: “105” -> 10cells/mL?

Response: It is changed now

Line 242-244: to be removed 

Response: It is changed now

Fig 1, Fig 2 and Table 1 legend: “lemon balm essential oil”? or Melissa officinalis essential oil?

Response: It is changed now to Melissa officinalis essential oil

Line 272- 360: From line 272 until line 360, none of the fungi name or even Melissa officinalis are in italic as they should.

Response: It is changed now

Figure 3,4,5: Y axis: the assays performed did not quantify a growth %, but the removal of a biofilm, measured by the remaining biofilm metabolic activity, expressed in % in relation to control.

The increased of the metabolic activity of the biofilms depending on the concentration of the antifungal compound or EO are difficult to understand. Did this happen in each replicate? The material and methods mentioned triplicates, but did the authors performed triplicates of 3 independent assays? If so, why did not they represent error bars?

Response: The y-axis's title is now "metabolic activity (%)." Statistical analysis is provided now, and significant results are appropriately marked.

Paragraph line 333: The synergy analysis was interpreted by Bliss analysis. The justification for this approach is explained in the discussion and an example of a correlation with the FICI determination is presented in the discussion. Could the authors include in supplementary results a table with the FICI of all the combinations used? So a reader could compare more easily with the literature? Moreover, not all the readers are able to understand the Figure 6. Moreover, the image quality must be increased. In this pdf version, I was not able to read the legends of each combination.

Response: Since other reviewers found this part of the discussion questionable, we have decided to exclude any comparison between the two methods and to present and explain only the Bliss analysis results. The quality of Figure 6 quality is improved now.

Line 361: Discussion

The discussion is very large, for example, line 384, there is no need to repeat the results. Although, the discussion is large, it is also very well written, present a good interpretation of the results and answers to questions that the readers may had along the paper.

Response: The discussion is reduced now

Line 447: I agree with the discussion but we also have to be aware that the effect of antifungals in the biofilms also depends on their mechanism of action. The nystatin exhibited a good antibiofilm activity because it binds the ergosterol already present in the fungal cells of the biofilm, leading to the fungal death. Yet, the fluconazole target enzymes of the synthesis of ergosterol. So it targets the de novo synthesis of ergosterol, with no effect on the cells already present in the biofilm.

Response: It is added to discussion (lines 481-485)

Line 517: I do not understand how the FICI calculation do not provide further information about effects at lower concentrations. No reference is cited to support this. For a reader, FICI calculation can be run with all the concentrations tested in the checkerboard assay as well. 

Response: FICI calculations imply taking only one concentration of each agent from the plate and cannot be performed for all combinations in the plate. Also, we performed FICI scoring for each combination of the tested agents and obtained results only for a few cases. This resulted from the lack of growth below 90% of control (reduction of 90%) or, in some cases, below even 50% (reduction of 50% of control). Therefore, we decided to include modeling, which would give extensive results, especially because it gives results that cannot be read using the FICI scoring. In order not to confuse the readers, we have decided to exclude this part of the discussion and to present and discuss only the results of the Bliss analysis.

Comment: The authors presented the Fig 3, 4 and 5 instead of determination of a value of SMIC because it was not possible to eradicate all the biofilm at the concentrations tested? Why did not they increase the concentration above 2x MIC?

Response: Higher concentrations were not tested because we could not properly dissolve the tested agents in these high concentrations. The method includes stock solutions dissolved in the plate ten times so the initial stocks were saturated.

Round 2

Reviewer 1 Report

In this revised version of the manuscript the authors have addressed most of my comments. However, the paragraph 2.5 in Material and methods section must be improved. As I have already suggested in my previous comments, in the subsection 2.5.1, in addition to the method used to quantify the biofilm formation, the authors should also include the treatment protocols used to evaluate the impact of FLU, NY and MOEO on preformed biofilm by Candida spp.. In the subsection 2.5.2 it should be described the method used to evaluate the impact of FLU, NY and MOEO on the metabolic activity of Candida biofilm (MTT assay), specifying that FLU, NY and MOEO treatments are reported in 2.5.1..

Minor editing of English language

Author Response

In this revised version of the manuscript the authors have addressed most of my comments. However, the paragraph 2.5 in Material and methods section must be improved. As I have already suggested in my previous comments, in the subsection 2.5.1, in addition to the method used to quantify the biofilm formation, the authors should also include the treatment protocols used to evaluate the impact of FLU, NY and MOEO on preformed biofilm by Candida spp.. In the subsection 2.5.2 it should be described the method used to evaluate the impact of FLU, NY and MOEO on the metabolic activity of Candida biofilm (MTT assay), specifying that FLU, NY and MOEO treatments are reported in 2.5.1.

Dear reviewer,

We clarified the part in the methodology concerning the applied methods we used for 2.5.1. biofilm production and in part 2.5.2. the method we used to examine the reduction of metabolically active cells by the effect of fluconazole, nystatin, and essential oil of Melissa officinalis essential oil. 

Sentence: The activities of each agent on the formed biofilm were tested for eight strains, four C. albicans and four C. glabrata isolates selected by their highest resistance rate to FLU, NY, and MOEO and high biofilm producing abilities

is now changed to

The activities of each agent on the formed biofilm were tested using MTT (3-(4,5-dimethyl-2-thiazolyl)-2,5-diphenyl-2H tetrazolium bromide) assay. Eight strains, four C. albicans and four C. glabrata isolates selected by their highest resistance rate to FLU, NY, and MOEO, and high biofilm-producing abilities were examined.

The English language is maximally improved.

Reviewer 2 Report

Thank you for addressing my comments, the manuscript has benefitted from the changes.

The paper is well written; routine copy editing should be sufficient to produce an impeccable manuscript.

Author Response

Dear reviewer, thank you very much for your suggestions and comments. Best regards